# Improving Corrosion and Wear Resistance of 316L Stainless Steel via In Situ Pure Ti and Ti6Al4V Coatings: Tribocorrosion and Electrochemical Analysis

**DOI:** 10.3390/ma18030553

**Published:** 2025-01-25

**Authors:** Darya Alontseva, Hasan İsmail Yavuz, Bagdat Azamatov, Fuad Khoshnaw, Yuliya Safarova (Yantsen), Dmitriy Dogadkin, Egemen Avcu, Ridvan Yamanoglu

**Affiliations:** 1School of Digital Technologies and Artificial Intelligence, D. Serikbayev East Kazakhstan Technical University, 19 Serikbayev Street, Ust-Kamenogorsk 070010, Kazakhstan; yantsen@nu.edu.kz; 2Department of Metallurgical and Materials Engineering, College of Engineering, Kocaeli University, Kocaeli 41001, Turkey; hasanismail.yavuz@kocaeli.edu.tr (H.İ.Y.); ryamanoglu@kocaeli.edu.tr (R.Y.); 3Smart Engineering Competence Centre, D. Serikbayev East Kazakhstan Technical University, 19 Serikbayev Street, Ust-Kamenogorsk 070010, Kazakhstan; bazamatov@ektu.kz (B.A.); ddogadkin@ektu.kz (D.D.); 4School of Engineering and Sustainable Development, De Montfort University, Leicester LE1 9BH, UK; fuad.khoshnaw@dmu.ac.uk; 5Laboratory of Bioengineering and Regenerative Medicine, National Laboratory Astana, Nazarbayev University, Astana 010000, Kazakhstan; 6Ford Otosan Ihsaniye Automotive Vocational School, Kocaeli University, Kocaeli 41650, Turkey; egemen.avcu@kocaeli.edu.tr; 7Department of Mechanical Engineering, Kocaeli University, Kocaeli 41001, Turkey

**Keywords:** biomaterials, corrosion, hot press, implant, stainless steel, titanium, wear

## Abstract

This study aims to achieve in situ-formed pure Ti and Ti6Al4V coatings on 316L stainless steel through hot pressing and examine their wear and corrosion properties thoroughly in two simulated body fluids: physiological serum (0.9% NaCl) and Hanks’ solution. The sintering and diffusion bonding process was conducted at 1050 °C under a uniaxial pressure of 40 MPa for 30 min in a vacuum environment of 10^−4^ mbar. Following sintering, in situ-formed pure Ti and Ti6Al4V coatings, approximately 1000 µm thick, were produced on 316L substrates approximately 3000 µm in thickness. The mean hardness of 316L substrates, pure Ti, and Ti6Al4V coatings are around 165 HV, 170 HV, and 420 HV, respectively. The interface of the stainless steel substrate and the pure Ti and Ti6Al4V coatings exhibited no microstructural defects, while the interface exhibited significantly higher hardness values (ranging from 600 to 700 HV). The coatings improved corrosion resistance in both electrolytes compared to the 316L substrate. Wet wear tests revealed reduced friction coefficients in 0.9% NaCl relative to Hanks’ solution, highlighting the chemical interactions between the material surface and the electrolyte type and the significance of tribocorrosion in biocoatings.

## 1. Introduction

Ti-based materials, austenitic stainless steel (316L), and cobalt-chromium (CoCr) alloys dominate the metallic implant market due to their excellent mechanical properties, corrosion resistance, and biocompatibility [1]. Among these materials, 316L stainless steel (316L) is widely used in dental, orthopedic, cardiovascular, and fixation devices owing to its high strength, ductility, machinability, and cost-effectiveness [2]. However, compared to other metallic biomaterials, 316L exhibits lower corrosion resistance and biocompatibility [3]. Corrosion is a critical issue affecting implants’ biocompatibility and structural integrity. The passive film structure on the material’s surface and the applied load control the surface degradation mechanisms. Additionally, the corrosive nature of body fluids creates extra stress on the surface, rendering the implant susceptible to tribocorrosion [4].

Tribocorrosion, characterized by concurrent wear and corrosion processes, refers to simultaneous mechanical and electrochemical interactions on the material’s surface [5]. Low hardness (160 HV) of 316L, particularly in load-bearing applications, results in significant wear [6]. Overextended use and corrosion lead to the release of metal ions such as Ni, Fe, and Cr into the body. These ions can cause blood clotting, restenosis in blood vessels, and fibrous tissue injuries in surrounding tissues [7,8,9]. Studies addressing these challenges focus on preventing the direct contact of 316L with body fluids. Research efforts are directed toward developing coatings with superior biocompatibility and corrosion resistance for the 316L implant surface [10,11,12,13,14]. The composition of the passive film formed on the surface varies depending on the material. Enhancing the stability of the oxide layer inhibits the release of ions into body fluids [15,16]. Consequently, the implant surface can be coated with materials possessing a better passive film layer to improve surface properties. As a solution, 316L’s surface is often coated with pure Ti or Ti alloys [17,18,19]. While 316L forms Cr_2_O_3_ as the passive film, titanium alloys predominantly form TiO_2_ along with other oxides, depending on the alloy type [20]. Elements such as Nb, Ta, and Mo, alloyed with Ti, form oxides like Nb_2_O_5_, Ta_2_O_5_, and MoO_3_ in addition to TiO_2_. These oxides enhance the stability of the passive film layer in corrosive environments [21,22,23,24,25]. Among these oxides, TiO_2_ is highly stable and inert under physiological and electrochemical conditions [26]. According to ellipsometry and XPS analyses, the passive film thickness is 2 nm for 316L and 9 nm for the Ti6Al4V alloy [27,28]. Increased passive film thickness correlates with enhanced corrosion resistance. On the other hand, diamond-like carbon (DLC) and TiN have emerged in recent years as potential materials for biomaterials due to their high hardness, low friction coefficient, excellent wear and corrosion resistance, chemical inertness, and high electrical resistance [29,30]. However, significant debates persist regarding the effectiveness of these coatings for biomedical applications. For instance, clinical trials of DLC-coated vascular stents have revealed that DLC coatings do not provide a substantial improvement in restenosis rates compared to uncoated stents [31]. Furthermore, the production processes for coatings like TiN and DLC are typically more complex and expensive [32]. Lastly, a ten-year follow-up study of DLC-coated artificial hip joints demonstrated that the failure rate of DLC-coated Ti6Al4V femoral heads was significantly higher than that of alumina femoral heads [33]. Therefore, although TiN and DLC coatings are harder and more wear-resistant, the inherent properties of pure Ti and Ti6Al4V make them more suitable for biocompatible and long-lasting implants.

Physical vapor deposition (PVD) and chemical vapor deposition (CVD) coatings are frequently implemented to enhance the corrosion and wear resistance of stainless steel [34]. However, factors such as target overheating, arc point movement, deposition rate, and pre-tension structure can lead to defects like particles and voids on the deposited coating surface [35,36]. These defects can grow due to variations in ion energy levels, lowering the success rate of coatings. Alternatively, hot pressing can be utilized to obtain in situ-formed coatings through simultaneously sintering metal powders (e.g., steel and titanium alloy powders). The processing of thick coatings through hot pressing provides significant advantages over conventional coating methods like PVD and CVD by eliminating porosity and ensuring homogeneity under vacuum conditions, which reduces equipment and operational costs. Unlike PVD and CVD, which often result in defects like porosities and delamination due to incomplete coatings, hot pressing achieves full surface bonding through uniaxial pressure, improving mechanical integrity [37,38]. Moreover, the reduced energy requirements of hot pressing make it a more sustainable and scalable option for biomedical applications. Studies demonstrated that coatings produced via hot pressing exhibit superior hardness and adhesion compared to PVD coatings while achieving comparable or better corrosion resistance [10,39].

The present study aims to simultaneously sinter 316L stainless steel with pure Ti and Ti6Al4V alloy powders through hot pressing to produce pure Ti and Ti6Al4V coatings on stainless steel substrates. The proposed processing strategy can be achieved at relatively low temperatures and within shorter processing times [40], while pressure applied during sintering allows the production of coatings with high density and superior mechanical properties. The wear and corrosion properties of the in situ-produced pure Ti- and Ti6Al4V-coated stainless steel samples were thoroughly investigated in two simulated body fluids: physiological serum (0.9% NaCl) and Hanks’ solution. The results given in this study constitute the initial part of the associated research. For a better understanding of electrochemical behavior, electrochemical impedance spectroscopy (EIS) and research across a wider polarization range will be used to study passive layer characteristics like charge transfer resistance and capacitance, which are crucial for assessing the long-term performance of the developed coatings under physiological conditions.

## 2. Materials and Methods

### 2.1. Materials

Pure Ti (−25 µm) grade 1 and 316L stainless steel (−63 µm) powders were supplied from Alfa Aesar (Haverhill, Massachusetts, USA). Ti6Al4V alloy (−43 µm) powders were obtained from Whole Metal (Elmhurst, Illinois, USA). Figure 1a,b present SEM images of the irregularly shaped 316L and pure titanium powders, respectively, while Figure 1c shows the spherical Ti6Al4V powders. A schematic view of the graphite mold, substrate, and coating layers is given in Figure 1d.

Before selecting production parameters, preliminary production studies were conducted to optimize the processing conditions. Temperature optimization was a primary focus in these studies, while the applied pressure was determined based on the load-bearing capacity of graphite dies during sintering. The optimum parameters for coating quality were identified as 1050 °C and 40 MPa. Temperatures below 1050 °C resulted in insufficient bonding between the coating and the substrate, with pores detected at the interface. Consequently, the samples were produced at 1050 °C under a constant uniaxial pressure of 40 MPa. Simultaneous sintering of 316L stainless steel, pure Ti, and Ti6Al4V alloy powders was performed using a DIEX VS50 pressure-assisted sintering system (Shanghai, China) in a 10^−4^ mbar vacuum atmosphere to prevent oxidation. Following sintering, in situ-formed pure Ti and Ti6Al4V coatings, approximately 1000 µm thick, were produced on 316L substrates approximately 3000 µm in thickness. Cylindrical samples with a diameter of 40 mm and a thickness of 4 mm were ultimately produced.

### 2.2. Microstructure and Hardness Characterization

Samples were sectioned with a Metkon 150 micro-cut precision cutting machine and molded with Mikrotest hot-molding equipment. Subsequently, the molded samples were ground with SiC papers (Akasel, Roskilde, Denmark) of increasing grit size (320, 600, 1000, and 2000) and polished with diamond suspensions (Akasel, Denmark) of 9 and 3 µm. The stainless steel substrate was etched with glycerol king water (20 mL glycerol, 50 mL HCl, 15 mL HNO_3_), while the coatings were etched with Kroll’s reagent (2 mL HF, 6 mL HNO_3_, 92 mL H_2_O). Finally, optical microscopy investigations of the etched samples were conducted using an Olympus BX41M-LED microscope (Olympus corporation, Tokyo, Japan). Hardness measurements were performed using a Future Tech EV-700 Vickers hardness tester (Future Tech corporation, Tokyo, Japan) with a 5 kg load applied for 15 s.

### 2.3. Tribological and Corrosion Characterization

The tribological properties of the samples were evaluated using a ball-on-disc-type TURKYUS POD/HT/WT wear testing device (Bursa, Turkey). Wet wear tests were performed at a 250 m sliding distance, 20 N load, and 55 mm/s rotational speed (107 rpm) at room temperature using a 6 mm stainless steel ball. Tests were conducted separately in physiological serum (0.9% NaCl) and Hanks’ balanced salt solution with pH = 7.25 ± 0.15 [composition: NaCl: 8 g/L, CaCl_2_: 0.14 g/L, KCl: 0.4 g/L, NaHCO_3_: 0.35 g/L, Na_2_HPO_4_ (anhydrous): 0.0477 g/L, KH_2_PO_4_: 0.06 g/L, MgSO_4_: 0.0977 g/L, and d-glucose: 1 g/L] [41]. The samples were kept in the solution for 30 min before the wet-sliding-wear test. Figure 2 displays the layout of the sliding-wear tester alongside an image of the test sample in Hanks’ solution.

Tafel extrapolation, linear polarization resistance, and open circuit potential (OCP) measurements were conducted using an AMETEK VersaSTAT 4 potentiostat/galvanostat in a three-electrode cell. The samples served as the working electrode, while platinum was the reference electrode. Before polarization testing, samples were immersed in 0.9% NaCl and Hanks’ solution for 30 min to reach equilibrium. Tafel extrapolation was employed to determine the corrosion current density (I_corr_) and corrosion potential (E_corr_). Polarization curves were obtained in 0.9% NaCl and Hanks’ solution at 24 °C, scanning the potential from −250 mV to +250 mV vs. SCE at a rate of 1 mV/s. The exposed sample area in the solutions was 1 cm² [42]. The polarization range utilized in this study (−250 mV to +250 mV vs. SCE) was selected to reflect the conditions most relevant to the biological environments in which the materials are expected to perform. This range was specifically chosen to evaluate the passive film formation and initial corrosion resistance behavior under simulated conditions without risking sample integrity. Subsequent to the polarization test, the surface alterations and corrosion mechanisms of the samples were examined utilizing electron microscopy.

## 3. Results and Discussion

### 3.1. Microstructural Characterization

Figure 3 shows the etched cross-sectional optical microscope images of the samples. Figure 3a shows the typical austenitic microstructure of the substrate material, 316L stainless steel, while α-Ti grains are prominent in the coating layer. Figure 3b shows the coating layer of the Ti6Al4V alloy with an α + β phase structure. Ti6Al4V alloy is characterized by HCP α grains with equiaxed morphology, surrounded by BCC β phases distributed around the α grains. The vanadium-rich β phase appears as light-colored lamellar regions, whereas the dark-colored equiaxed grains represent the α phase-rich areas. In both images, a distinct interfacial layer formed due to diffusion between the substrate and coating materials is evident. The diffusion of titanium and aluminum elements into the 316L stainless steel substrate may form this thin interfacial layer, marked in Figure 3a,b. Ti and Al interact with Fe, Cr, and Ni elements during the coating process, forming various phases. As a result of the diffusion effect, a material interlayer containing varying amounts of Ti and Fe at different depths may be formed. The differing diffusion rates of Ti and Fe into each other can trigger the formation of intermetallic Ti_x_Fe_y_ compounds. Fe and Ni diffuse rapidly and migrate into the titanium region, causing the formation of various compounds at the interface [43]. Song et al. [44] reported that TiNi, Ti_2_Ni, and Fe_2_Ti compounds were identified at the interface between 316L stainless steel and TiN coatings deposited using the chemical vapor deposition method. Similar studies have also supported the formation of FeTi and Fe_2_Ti intermetallics [40,45].

Additionally, no cracks or porosities were observed in the interfacial region. The absence of such defects ensures a strong bond layer, enhancing the strength and toughness of the interfacial region. Pure Ti- and Ti6Al4V-coated stainless steel samples (316L/pure Ti and 316L/Ti6Al4V samples) exhibited homogeneously distributed microstructural features, yielding a high density.

### 3.2. Microhardness Results

Figure 4 presents the hardness analysis results for the substrate, interface, and coatings. In the 316L/pure Ti sample, the average hardness value of the coating was measured as 170 HV. The hardness increased toward the interface, reaching an average value of 600 HV in the diffusion zone. Conversely, the hardness gradually decreased away from the interface, with the 316L substrate’s hardness measured as 165 HV. For the 316L/Ti6Al4V sample, the average hardness of the coating was 420 HV. An increase in hardness was observed at the interface, yielding similar results to the other sample, with an average interface hardness of 710 HV. The hardness of the substrate material measured approximately 170 HV. The highest hardness values in all samples were observed at the interface, aligning with findings in the literature. For example, Auger et al. [45] produced Fe/Ti, 316L/Ti, and 316L/sputtered Ti coatings using the magnetron sputtering method. TEM and SEM images and EDS analyses at the interface showed changes in the titanium microstructure due to Fe diffusion. Interface analyses revealed the formation of FeTi and Fe_2_Ti intermetallic compounds. FeTi exhibits a BCC lattice structure with lattice parameters close to α-Fe and is known for its hardness, wear resistance, and brittleness. Fe_2_Ti has a Laves-type hexagonal close-packed crystal structure and is also very hard (1242 HV0.5) and brittle [46]. The presence of these compounds at the interface increased the hardness compared to the coating and substrate. Similarly, Wathanyu et al. [3] coated 316L substrates with pure Ti and TiN using the PVD method. The hardness of 316L coated with pure Ti and TiN increased by 50% and 85%, respectively.

In the present study, the interface hardness of the 316L/pure Ti sample was measured at 620 HV, consistent with the values reported in the literature. Significantly higher hardness values (e.g., 1950 HV) were previously reported at the interface between 316L and Ti-based coating samples, depending on the coating’s composition [47]. In another study, Malkiya Rasalin Prince et al. [2] coated 316L substrates with a Ti6Al4V-3ZrC composite using the magnetron sputtering method and reported an interface hardness of 1290 HV. Hardness analysis verified the microstructural characteristics at the interface between 316L and Ti-based materials, as described in Section 3.1. In other words, the interface hardness of Ti-based coatings on 316L increased due to the formation of intermetallics.

### 3.3. Wet-Sliding-Wear Test Results

Figure 5 shows the coefficient of friction (COF) curves obtained from wet-sliding-wear tests conducted in simulated body fluid (SBF) using Hanks’ solution and physiological serum. The analysis of COF values reveals that all materials exhibited higher COF values in Hanks’ solution. In 0.9% NaCl, the COF values for the 316L substrate, 316L/Ti, and 316L/Ti6Al4V-coated samples were measured as 0.20 ± 0.08, 0.35 ± 0.07, and 0.20 ± 0.12, respectively. Conversely, in Hanks’ solution, all materials displayed similar COF values, averaging around 0.4. In 0.9% NaCl solution, chloride ions stabilize the passive oxide layer by forming a protective barrier that mitigates surface degradation [48]. This stabilization reduces the CoF by limiting active material removal and preserving the surface integrity during sliding. Conversely, in Hanks’ solution, the presence of phosphate and bicarbonate ions induces a competing interaction with the surface oxide layer. Phosphate ions can form weak complexes with metal oxides, while bicarbonate ions promote localized pH shifts, potentially destabilizing the oxide layer [49]. This destabilization leads to increased wear and higher friction coefficients. The tribochemical interactions are influenced by the ionic strength and concentration gradients, which alter the electrochemical potential of the oxide layer [50]. These results underscore the intricate interplay between the ionic composition of the surrounding environment and the mechanical and chemical processes driving tribocorrosive wear. The COF values initially increased sharply during the first few cycles but subsequently decreased and stabilized after a sliding distance of 50 m. Notably, the COF of 316L in Hanks’ solution exhibited a sharp initial increase, followed by a gradual decrease throughout the test. These fluctuations can be attributed to the formation and detachment of wear debris [51].

As stated in Section 2.3, samples were immersed in the solutions for 30 min before the wear tests to achieve equilibrium between the material and the solution. Variations in solution composition influenced the surface properties of the materials, resulting in changes in COF values. However, after reaching equilibrium, COF values remained stable for all samples. A similar study by Pandey et al. [52] investigated the wear behavior of biomedical-grade 316L, pure Ti, and Ti6Al4V alloys in artificial saliva (AS), simulated body fluid (SBF), Ringer’s solution (RS), and phosphate-buffered saline (PBS). Their results showed that COF values varied depending on the solution. For instance, the lowest COF value for SS 316L was 0.44 ± 0.05 in PBS, while pure Ti was 0.49 ± 0.06 in Ringer’s solution. Similarly, the lowest COF value for the Ti6Al4V sample was 0.42 ± 0.03 in RS. The average COF values also increased depending on the solution, with 316L reaching up to 0.74 ± 0.09 in RS, pure Ti showing 0.87 ± 0.06 in AS, and Ti6Al4V exhibiting 0.92 ± 0.11 in SBF. Miura-Fujiwara et al. [53] conducted wear tests on pure Ti and SS 316L in AS and deionized water. Their findings indicated an average COF of 0.6 in AS and 0.45 in water for 316L. The average COF remained unchanged at 0.45 across both solutions for pure Ti. Ruggiero et al. [54] investigated the bio-tribocorrosive behavior of Ti6Al4V and 316L in RS and physiological serum using a tribometer equipped with a three-electrode potentiostat. They observed the highest COF values for 316L in both solutions. The lower COF values measured for the 316L/Ti6Al4V alloy were attributed to its higher hardness. COF values for both samples significantly decreased when the solution was changed. Experiments revealed that factors like pH and higher NaCl concentrations resulted in greater material loss due to the synergistic mechanical–chemical effects, highlighting the multidisciplinary nature of the tribocorrosion phenomenon, where both mechanical and chemical properties of the material play a significant role. Chen et al. [51] studied the tribocorrosion behavior of Ti6Al4V and AISI 316 stainless steel in seawater (3.5% NaCl) and deionized water. Using a 100 N load and an Al_2_O_3_ abrasive ball, they measured COF values of 0.6 and 0.5 in seawater and deionized water, respectively, for stainless steel. For the Ti6Al4V alloy, COF values were 0.3 in seawater and 0.2 in deionized water, indicating that stainless steel exhibited higher COF values than Ti6Al4V. Additionally, their findings suggested higher COF values in seawater than in deionized water, emphasizing seawater’s anti-friction effects. In light of these observations, the presented COF curves obtained under different solutions (Figure 5) are consistent with the literature, with COF values varying according to the solution used.

### 3.4. Electrochemical Analysis

Material corrosion transfers metal ions into the body, leading to serious health risks. Additionally, corrosion-induced implant loosening requires revision operations [55]. It is well-established that the success rate of re-implantation is lower than that of the initial implantation [56]. Moreover, revision surgeries prolong the patient’s recovery and increase the pain experienced [57,58]. Due to bodily fluids’ aggressive and corrosive character, electrochemical properties are among the most critical parameters. Therefore, the present study performed electrochemical tests on the substrate and coating materials through open circuit potential testing (OCP testing). OCP refers to the voltage between the working and reference electrodes when no potential or current is applied to the cell. In a reversible electrode system, OCP is also known as the equilibrium potential [21]. Before polarization testing, the OCP test was conducted on the substrate and coating materials for 30 min in 0.9% NaCl and Hanks’ solution.

Figure 6a presents the OCP test results of the samples in 0.9% NaCl solution. The diagram indicates that the corrosion potential curve for 316L/Ti started at −130 mV and stabilized at −215 mV after 1800 s. The curve remained stable for the first 500 s, followed by fluctuations until the oxide layer achieved stability. For the 316L/Ti6Al4V sample, the corrosion potential, which started at −255 mV, reached −335 mV after 1800 s. In contrast, the corrosion potential for the 316L sample, initially measured as −310 mV, converged to the same value as 316L/Ti6Al4V by the end of the test. The higher potential of 316L/Ti compared to the other sample indicates the formation of a more noble oxide layer [59]. Unlike the 316L/Ti sample, no peak formation was observed in the OCP curves for the 316L/Ti6Al4V and 316L samples, suggesting that their oxide film layers stabilized more quickly than pure Ti.

Figure 6b shows the OCP test results in Hanks’ solution. Similar values to those in 0.9% NaCl solution were observed for 316L and 316L/Ti6Al4V samples. However, the curve for 316L/Ti demonstrated greater stability in Hanks’ solution, with the corrosion potential starting at −350 mV and stabilizing at −410 mV by the end of the test. The potential behavior of the samples varies due to the chemical nature of the environment [54]. Indeed, the slightly lower pH of 0.9% NaCl compared to Hanks’ solution and different salt compositions caused variations in the OCP diagrams [60]. In a study by Carquigny et al. [59], OCP tests and various polarization tests were performed on Ti6Al4V alloy and 316L stainless steel in 0.9% NaCl, Ringer’s solution, PBS, and bovine serum albumin (BSA) solutions. The results showed that the corrosion potential of the Ti6Al4V alloy ranged from −290 mV to −325 mV in all solutions. For the 316L alloy, the potential values varied more significantly depending on the solution, ranging from −100 mV to −250 mV. Similar results were obtained for the 316L/Ti6Al4V sample in this study. However, higher potential values were measured for the 316L sample in 0.9% NaCl compared to the present study.

Following the OCP test, the samples’ I_corr_ and E_corr_ were measured using Tafel extrapolation. The potential range applied in this test plays a crucial role in modeling the corrosive behavior of actual biological applications, as it simulates the chemical changes in body fluids resulting from dietary habits [54]. This study conducted I_corr_ measurements at E_corr_ values ranging from −250 mV to +250 mV. The Tafel curves obtained in both solutions are shown in Figure 7. The I_corr_ and E_corr_ values derived from curve extrapolation and the linear polarization resistance (R_p_) are provided in Table 1. The corrosion potential in the diagram represents the potential of the corroding surface in an electrolyte relative to a reference electrode. This value is determined by Tafel’s extrapolation of the anodic and cathodic curves in a potentiodynamic polarization diagram. Similarly, the corrosion current density is measured using potentiodynamic polarization curves and is strongly related to the corrosion rate. Better corrosion performance of a material is achieved by increasing E_corr_ and reducing I_corr_ [7,61]. Polarization resistance results are determined by taking the slope of the region considered linear on the potential–current curve (E_corr_ ± 10 mV). R_p_ indicates resistance to dissolution and is inversely proportional to I_corr_ [62].

As seen in Figure 7a and Table 1, 316L/pure Ti showed a higher corrosion potential and lower corrosion current density than 316L/Ti6Al4V and 316L in 0.9% NaCl. Additionally, the measured E_corr_ value was consistent with the OCP test. On the other hand, 316L/Ti6Al4V showed a lower I_corr_ and higher E_corr_ value than 316L. Based on these observations, 316L/pure Ti demonstrated the highest corrosion performance in 0.9% NaCl, whereas 316L performed the worst among the materials. Examination of the test results in Hanks’ solution (Figure 7b) showed similar I_corr_ values for 316L/Ti6Al4V and 316L/pure Ti; however, 316L/Ti6Al4V exhibited a higher E_corr_ value. Conversely, a lower E_corr_ value was observed for 316L/pure Ti in Hanks’ solution compared to 0.9% NaCl. A similar trend was observed for 316L/pure Ti in the OCP test.

In Hanks’ solution, the buffering effect and the presence of multivalent ions, such as phosphate and bicarbonate, introduce complex interactions with the passive oxide layer [63]. Phosphate ions can compete with chloride ions, altering the adsorption and dissolution dynamics at the metal interface. Bicarbonate ions further modify the local pH and potentially destabilize the oxide layer, increasing wear rates and creating a higher likelihood of localized corrosion [64]. In contrast, 0.9% NaCl, with its relatively simple ionic composition, primarily facilitates chloride-induced breakdown of the passive layer. However, this simpler environment allows for a more stable oxide layer under lower mechanical stresses, resulting in comparatively lower wear and corrosion rates. Differences in mechanical and electrochemical synergies between the two solutions play a significant role. In Hanks’ solution, the simultaneous mechanical action and chemical reactivity exacerbate tribocorrosive wear, particularly in the Ti6Al4V coating, due to its sensitivity to multivalent ion interactions. The pure Ti coating demonstrates superior corrosion resistance in Hanks’ solution, potentially due to forming a more adherent and stable oxide layer. In 0.9% NaCl, both coatings exhibit comparable corrosion resistance, suggesting a dominant role of chloride ions in determining the passive film’s stability. These observations highlight the intricate interplay between chemical composition, pH buffering capacity, and tribological conditions in driving the material degradation processes in different electrolytes. Consequently, 316L/pure Ti and 316L/Ti6Al4V demonstrated higher corrosion resistance than 316L in Hanks’ solution. The polarization resistance results were also consistent with the I_corr_ results. The corrosion resistance of metallic biomaterials is determined by the structure of the passive film layer formed on the surface. Titanium naturally develops a highly protective oxide layer (TiO_2_) with a 2–5 nm thickness on its surface [55]. Furthermore, the repassivation rate of pure Ti is rapid due to its classification as a highly reactive metal [65]. Stainless steels are among the most popular metallic biomaterials due to their excellent corrosion resistance and mechanical properties. The corrosion resistance of AISI 316L stainless steel is primarily attributed to the spontaneous formation of chromium oxide (Cr_2_O_3_) film on its surface [51].

The oxide layer on pure titanium is typically thicker and more stable compared to Ti6Al4V. Under normal physiological conditions, it forms a stable TiO_2_ layer that can grow up to several nanometers in thickness, typically ranging from 1 to 10 nm [66,67]. This oxide layer forms immediately upon exposure to air or oxygen-containing environments and serves as a protective barrier against further oxidation. Generally, the oxide layer on pure titanium is continuous and uniform, providing excellent corrosion resistance. In contrast, the oxide layer on Ti6Al4V is usually thinner, ranging from 1 to 5 nm [68]. The presence of Al and V in the alloy influences the oxide structure. While a TiO_2_ layer still forms, it is often accompanied by additional oxide phases, such as Al_2_O_3_ and V_2_O_5_, which can affect the overall properties of the oxide layer [69]. However, the oxide layer on Ti6Al4V may not be as continuous as that on pure titanium. The alloy’s microstructure can lead to localized variations in the formation of the oxide layer. For example, Al contributes to a more compact and protective oxide layer, whereas vanadium may introduce discontinuities, increasing the potential for localized breakdown and making the oxide layer more prone to damage. Consequently, Ti6Al4V exhibits slightly lower corrosion resistance than pure titanium, especially in environments where localized breakdown of the oxide layer is likely. Additionally, the presence of V and Al in the oxide layer can contribute to galvanic corrosion when the oxide layer is disrupted due to vanadium’s higher electrochemical activity compared to titanium [70]. As a result, the electrochemical behavior of Ti6Al4V may demonstrate higher susceptibility to localized corrosion and lower long-term stability when compared to pure titanium. The polarization tests conducted within the range of −250 mV to +250 mV vs. SCE have provided valuable insights into the corrosion resistance and passive film stability of the developed coatings. It is significant to emphasize that extending the polarization range to elevated potentials may provide further insights into the transition from continuous anodic dissolution to passivation, along with the material’s behavior in more aggressive electrochemical conditions.

Following the polarization test, the samples’ surface changes and corrosion mechanisms were analyzed using SEM. Surface images of the samples are shown in Figure 8. As seen in Figure 8a, pits formed on the 316L surface as a result of corrosion after the Tafel test. Pitting corrosion is well-known as a characteristic type of 316L SS [71]. Pitting corrosion occurs as a result of the localized breakdown of the passive film layer on the metal surface in a corrosive environment. The most common pitting corrosion mechanism involves passive layer disruption followed by localized metal dissolution [72]. Thus, the pitting corrosion resistance of 316L primarily depends on the stability of the Cr_2_O_3_ layer covering its surface. Any microstructural feature, such as precipitates, grain boundaries, porosity, or dislocation structures that could compromise the integrity of this passive oxide film may lead to reduced corrosion resistance [73]. In contrast, the images of 316L/pure Ti and 316L/Ti6Al4V alloy shown in Figure 8b,c reveal no visible changes on their surfaces. In other words, no corrosion cells were formed on the surfaces of Ti-based samples after the Tafel test was conducted in both solutions. Based on these observations, it can be concluded that the obtained graphs are consistent with the microstructural findings.

## 4. Conclusions

This study employed the hot pressing method to produce pure Ti and Ti6Al4V alloy coatings on 316L stainless steel. The primary objective was to enhance the wear and electrochemical properties of 316L stainless steel. The hot pressing method successfully produced dense, defect-free coatings, improving performance metrics. The key findings of the study are presented as follows.

Hardness measurements revealed significant improvements at the coating and substrate interface, particularly for Ti6Al4V coatings. The interface of the stainless steel substrate and the pure Ti and Ti6Al4V coatings exhibited no microstructural defects, while the interface exhibited significantly higher hardness values. The 316L/Ti6Al4V alloy coating demonstrated higher hardness values at both interface and coating regions than 316L/pure Ti.

Electrochemical tests, including Tafel extrapolation and OCP measurements, demonstrated that pure Ti and Ti6Al4V coatings significantly enhanced the corrosion resistance of 316L stainless steel in both 0.9% NaCl and Hanks’ solution. Wet-sliding-wear tests revealed that the COF values of coated samples were generally lower in 0.9% NaCl solution compared to Hanks’ solution. This behavior underscores the role of the electrolyte’s chemical composition in influencing tribocorrosion mechanisms. Both coatings exhibited stable COF values.

Although 316L is corrosion-resistant, its low hardness makes it prone to wear, releasing ions into the body. Toxic elements such as Ni and Cr, as mentioned earlier, pose carcinogenic risks and endanger the patient’s life. In this context, using Ti-based materials as coatings in this study improved the corrosion resistance of the 316L substrate. The proposed innovative coating strategies may improve the 316L’s suitability for biomedical applications, particularly load-bearing implants. Future studies should be conducted to assess the biocompatibility and cytotoxicity of these coatings for clinical application.

The polarization range used in this investigation (−250 mV to +250 mV vs. SCE) was chosen to reflect the biological settings in which approximate passive film development and initial corrosion resistance without compromising sample integrity can be observed. However, increasing the polarization range to higher potentials could reveal more about the transition from continuous anodic dissolution to passivation and the material’s reaction in more aggressive electrochemical conditions. The findings presented in this study represent the preliminary phase of the associated research. Thus, future research will include an expanded potential range to better understand these coatings’ anodic behavior under extended electrochemical conditions.

This work predominantly utilized potentiodynamic polarization to assess the corrosion resistance and passive film stability of the coatings while recognizing the need of electrochemical impedance spectroscopy (EIS) for a more comprehensive understanding of the electrochemical behavior, providing insights into passive layer characteristics, such as charge transfer resistance and capacitance, which are critical in evaluating long-term performance under physiological conditions. The incorporation of EIS will be prioritized in future research to attain a more thorough assessment of the coatings’ performance.

## Figures and Tables

**Figure 1 materials-18-00553-f001:**
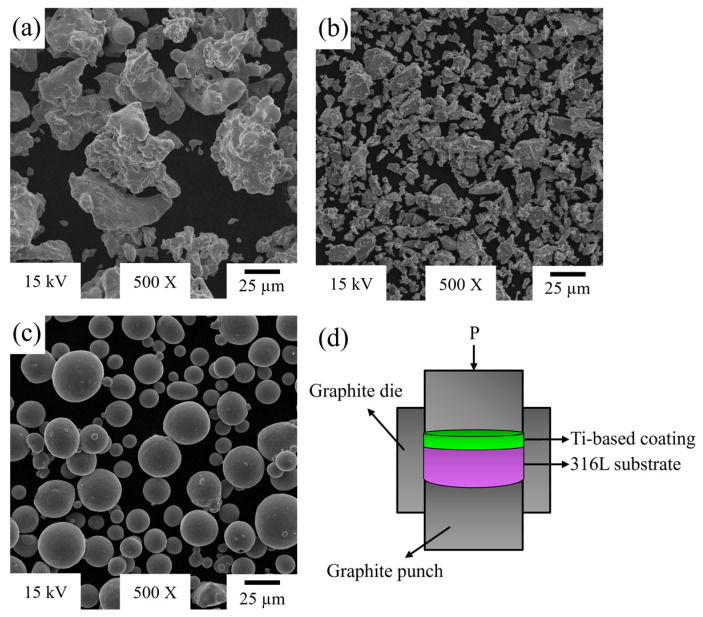
The SEM images of powders: (**a**) 316L, (**b**) pure Ti, (**c**) Ti6Al4V, (**d**) schematic representation of hot pressing setup.

**Figure 2 materials-18-00553-f002:**
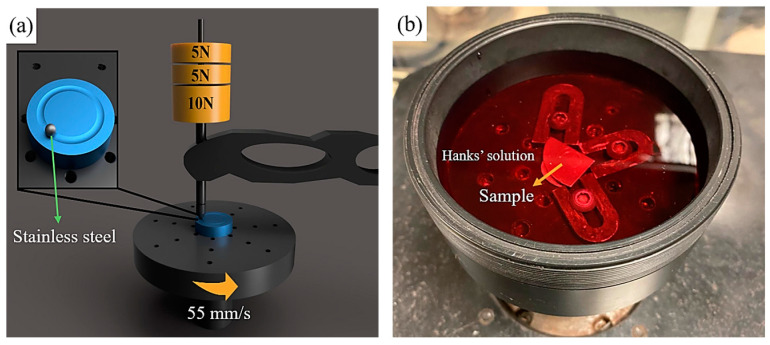
(**a**) Schematic representation of sliding-wear test setup, (**b**) image of test sample in Hanks’ solution for wet-sliding-wear test.

**Figure 3 materials-18-00553-f003:**
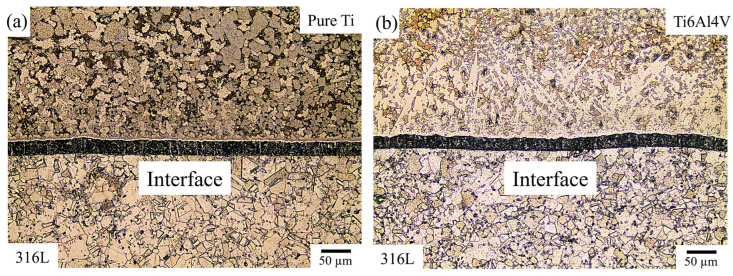
Optical microscope images of the etched samples: (**a**) 316L/pure Ti, (**b**) 316L/Ti6Al4V.

**Figure 4 materials-18-00553-f004:**
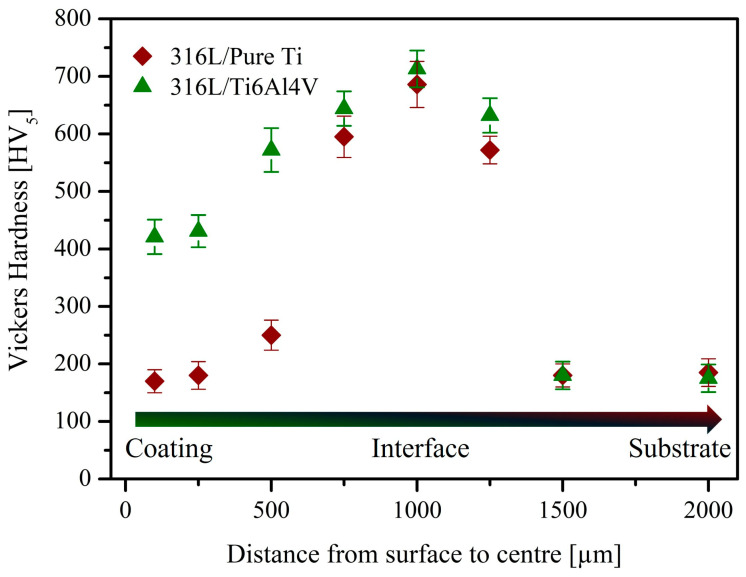
Cross-section Vickers hardness values of the samples as a function of distance from the surface, showing the variation of hardness in coating, interface, and substrate.

**Figure 5 materials-18-00553-f005:**
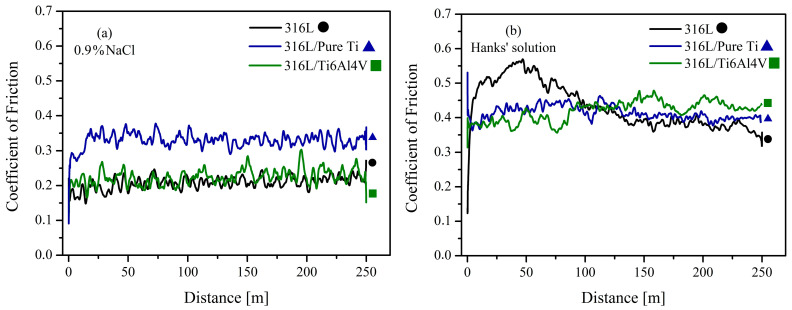
Coefficient of friction (COF) vs. distance during wet wear test in (**a**) 0.9% NaCl and (**b**) Hanks’ solution.

**Figure 6 materials-18-00553-f006:**
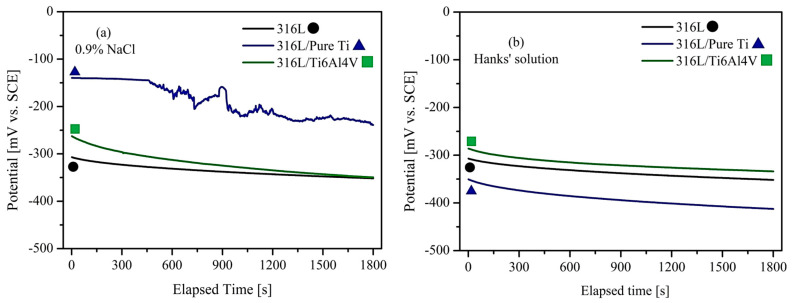
Comparative open circuit potential (OCP) values of 316L, 316L/pure Ti, and 316L/Ti6Al4V in (**a**) 0.9% NaCl and (**b**) Hanks’ solution.

**Figure 7 materials-18-00553-f007:**
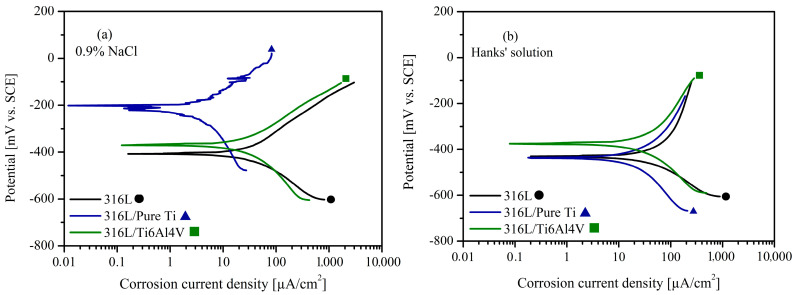
Tafel extrapolation curves of 316L, 316L/pure Ti, and 316L/Ti6Al4V in (**a**) 0.9% NaCl and (**b**) Hanks’ solution.

**Figure 8 materials-18-00553-f008:**
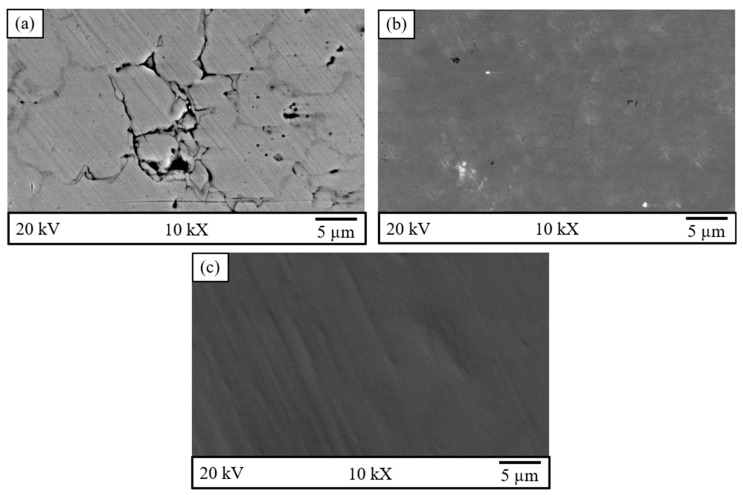
SEM image of sample surfaces after corrosion test: (**a**) 316L, (**b**) 316L/pure Ti (**c**) 316L/Ti6Al4V.

**Table 1 materials-18-00553-t001:** Tafel extrapolation and linear polarization resistance test results.

	%0.9 NaCl			Hanks’ Solution		
Sample	E_corr_ [mV]	I_corr_ [µA/cm^2^]	R_p_ [ohm]	E_corr_ [mV]	I_corr_ [µA/cm^2^]	R_p_ [ohm]
316L	−411	18.2	572	−430	16.3	712
316L/Pure Ti	−211	1.1	3289	−438	2.8	2589
316L/Ti6Al4V	−366	7.2	1256	−373	10.2	847

## Data Availability

Data are contained within the article. Further inquiries can be directed at the corresponding author due to institutional policy.

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
