# Peer review of "Improving Corrosion and Wear Resistance of 316L Stainless Steel via In Situ Pure Ti and Ti6Al4V Coatings: Tribocorrosion and Electrochemical Analysis"

_materials, 2025, doi:10.3390/ma18030553_

Round 1
Reviewer 1 Report
Comments and Suggestions for Authors
This paper provides a detailed experimental study on the corrosion and wear resistance of two coatings on 316L stainless steel. The experimental design is reasonable, and the analysis is comprehensive. After review, I believe the experimental analysis and conclusions are appropriate and hold certain academic value. Therefore, I recommend the paper for publication.
Author Response
Comment 1 This paper provides a detailed experimental study on the corrosion and wear resistance of two coatings on 316L stainless steel. The experimental design is reasonable, and the analysis is comprehensive. After review, I believe the experimental analysis and conclusions are appropriate and hold certain academic value. Therefore, I recommend the paper for publication.
Response:
Thanks for your kind comment and suggesting publication of our work.

Reviewer 2 Report
Comments and Suggestions for Authors
Dear Authors,
The paper presents a study on applying pure Ti and Ti6Al4V coatings on 316L stainless steel using the hot pressing method. The main objective is to improve the wear and corrosion resistance properties of 316L steel, a material widely used in biomedical implants. The research evaluates the tribological and electrochemical properties of the coatings in two simulated fluids: physiological saline solution (0.9% NaCl) and Hanks solution. The results show that the coatings significantly improve corrosion and wear resistance, highlighting the method's potential for biomedical applications.
However, some things could be improved: The results presented could be discussed in more depth to highlight their relevance; the data presented in the tables and figures could be detailed more, relating them directly to practical conclusions; and the Innovative Contribution could be clarified by Highlighting how the hot pressing method differs from other techniques, such as PVD and CVD, especially in terms of cost-effectiveness and efficiency.
However, the relationship between the tribological results and the chemical mechanisms involved is not explored in detail, and the specific influence of the solutions' chemical components on the tribochemical interaction has not been extensively discussed. Therefore, I recommend expanding the study to address how the chemical composition of the simulated solutions influences the tribocorrosive mechanisms.
The paper does not mention the application of EIS, having focused mainly on anodic polarization and open circuit potential (OCP) methods to evaluate corrosion resistance. These tests are adequate, but EIS would allow a deeper analysis of the properties of the passive layer and the corrosion mechanisms. Therefore, I recommend including EIS as a complementary method, as it provides a more detailed view of the electrochemical interface parameters, such as charge transfer resistance and capacitance of the passive layer, presenting Nyquist and Bode plots and discussing the evolution of the resistance and stability of the passive layer in different simulated solutions.
Author Response
The paper presents a study on applying pure Ti and Ti6Al4V coatings on 316L stainless steel using the hot-pressing method. The main objective is to improve the wear and corrosion resistance properties of 316L steel, a material widely used in biomedical implants. The research evaluates the tribological and electrochemical properties of the coatings in two simulated fluids: physiological saline solution (0.9% NaCl) and Hanks’ solution. The results show that the coatings significantly improve corrosion and wear resistance, highlighting the method's potential for biomedical applications. However, some things could be improved;
Response:
We thank you for addressing the required revisions. We have made major revisions throughout the manuscript following the reviewers’ comments and suggestions. We hope that, following these changes, the quality of our manuscript would conform to your standards.
Comment #1:
The results presented could be discussed in more depth to highlight their relevance; the data presented in the tables and figures could be detailed more, relating them directly to practical conclusions; and the Innovative Contribution could be clarified by Highlighting how the hot-pressing method differs from other techniques, such as PVD and CVD, especially in terms of cost-effectiveness and efficiency.
Response:
As requested, we have made substantial revisions throughout the manuscript. The revised manuscript now includes a more depth discussion of the results, significantly strengthening the impact of the paper. We have cited more than 20 new articles to further improve the discussion in the paper. Please see below some examples of the additions according to your suggestions.
L84-100:
Physical vapor deposition (PVD) and chemical vapor deposition (CVD) coatings are frequently implemented to enhance the corrosion and wear resistance of stainless steel [34]. However, factors such as target overheating, arc point movement, deposition rate, and pre-tension structure can lead to defects like particles and voids on the deposited coating surface [35, 36]. These defects can grow due to variations in ion energy levels, lowering the success rate of coatings. Alternatively, hot pressing can be utilized to obtain in-situ-formed coatings through simultaneously sintering metal powders (e.g., steel and titanium alloy powders). The processing of thick coatings through hot pressing provides significant advantages over conventional coating methods like PVD and CVD by eliminating porosity and ensuring homogeneity under vacuum conditions, which reduces equipment and operational costs. Unlike PVD and CVD, which often result in defects like porosities and delamination due to incomplete coatings, hot pressing achieves full surface bonding through uniaxial pressure, improving mechanical integrity [37, 38]. Moreover, the reduced energy requirements of hot pressing make it a more sustainable and scalable option for biomedical applications. Studies demonstrated that coatings produced via hot pressing exhibit superior hardness and adhesion compared to PVD coatings while achieving comparable or better corrosion resistance [10, 39].
L226-234:
In the present study, the interface hardness of the 316L/Pure Ti sample was measured at 620 HV, consistent with the values reported in the literature. Significantly higher hardness values (e.g., 1950 HV) were previously reported at the interface between 316L and Ti-based coating samples, depending on the coating’s composition [47]. In another study, Malkiya Rasalin Prince et al. [2] coated 316L substrates with a Ti6Al4V-3ZrC composite using the magnetron sputtering method and reported an interface hardness of 1290 HV. Hardness analysis verified the microstructural characteristics at the interface between 316L and Ti-based materials, as described in Section 3.1. In other words, the interface hardness of Ti-based coatings on 316L increased due to the formation of intermetallics.
L245-256:
In 0.9% NaCl solution, chloride ions stabilize the passive oxide layer by forming a protective barrier that mitigates surface degradation [48]. This stabilization reduces the CoF by limiting active material removal and preserving the surface integrity during sliding. Conversely, in Hanks' solution, the presence of phosphate and bicarbonate ions induces a competing interaction with the surface oxide layer. Phosphate ions can form weak complexes with metal oxides, while bicarbonate ions promote localized pH shifts, potentially destabilizing the oxide layer [49]. This destabilization leads to increased wear and higher friction coefficients. The tribochemical interactions are influ-enced by the ionic strength and concentration gradients, which alter the electrochemical potential of the oxide layer [50]. These results underscore the intricate interplay between the ionic composition of the surrounding environment and the mechanical and chemical processes driving tribocorrosive wear.
L368-386:
In Hanks' solution, the buffering effect and the presence of multivalent ions, such as phosphate and bicarbonate, introduce complex interactions with the passive oxide layer [63]. Phosphate ions can compete with chloride ions, altering the adsorption and dissolution dynamics at the metal interface. Bicarbonate ions further modify the local pH and potentially destabilize the oxide layer, increasing wear rates and a higher likelihood of localized corrosion [64]. In contrast, 0.9% NaCl, with its relatively simple ionic composition, primarily facilitates chloride-induced breakdown of the passive layer. However, this simpler environment allows for a more stable oxide layer under lower mechanical stresses, resulting in comparatively lower wear and corrosion rates. Differences in mechanical and electrochemical synergies between the two solutions play a significant role. In Hanks' solution, the simultaneous mechanical action and chemical reactivity exacerbate tribocorrosive wear, particularly in the Ti6Al4V coating, due to its sensitivity to multivalent ion interactions. The pure Ti coating demonstrates superior corrosion resistance in Hanks' solution, potentially due to forming a more adherent and stable oxide layer. In 0.9% NaCl, both coatings exhibit comparable corrosion resistance, suggesting a dominant role of chloride ions in determining the passive film's stability. These observations highlight the intricate interplay between chemical composition, pH buffering capacity, and tribological conditions in driving the material degradation processes in different electrolytes.
L396-422:
“The oxide layer on pure titanium is typically thicker and more stable compared to Ti6Al4V. Under normal physiological conditions, it forms a stable TiO₂ layer that can grow up to several nanometers in thickness, typically ranging from 1 to 10 nm [66, 67]. This oxide layer forms immediately upon exposure to air or oxygen-containing environments and serves as a protective barrier against further oxidation. Generally, the oxide layer on pure titanium is continuous and uniform, providing excellent corrosion resistance. In contrast, the oxide layer on Ti6Al4V is usually thinner, ranging from 1 to 5 nm [68]. the presence of Al and V in the alloy influences the oxide structure. While a TiO₂ layer still forms, it is often accompanied by additional oxide phases, such as Al₂O₃ and V₂O₅, which can affect the overall properties of the oxide layer [69]. However, the oxide layer on Ti6Al4V may not be as continuous as that on pure titanium. The alloy’s microstructure can lead to localized variations in the formation of the oxide layer. For example, Al contributes to a more compact and protective oxide layer, whereas vanadium may introduce discontinuities, increasing the potential for localized breakdown and making the oxide layer more prone to damage. Consequently, Ti6Al4V exhibits slightly lower corrosion resistance than pure titanium, especially in environments where localized breakdown of the oxide layer is likely. Additionally, the presence of V and Al in the oxide layer can contribute to galvanic corrosion when the oxide layer is disrupted due to vanadium’s higher electrochemical activity compared to titanium [70]. As a result, the electrochemical behavior of Ti6Al4V may demonstrate higher susceptibility to localized corrosion and lower long-term stability when compared to pure titanium. The polarization tests conducted within the range of -250 mV to +250 mV vs. SCE have provided valuable insights into the corrosion resistance and passive film stability of the developed coatings. It is significant to emphasize that ex-tending the polarization range to elevated potentials may provide further insights into the transition from continuous anodic dissolution to passivation, along with the mate-rial's behavior in more aggressive electrochemical conditions.”
Comment #2:
However, the relationship between the tribological results and the chemical mechanisms involved is not explored in detail, and the specific influence of the solutions' chemical components on the tribochemical interaction has not been extensively discussed. Therefore, I recommend expanding the study to address how the chemical composition of the simulated solutions influences the tribocorrosive mechanisms.
Response:
We have expanded the discussion in the Results section to explore the influence of chemical components in the simulated solutions on tribocorrosive mechanisms. For instance, please see the newly added discussion to emphasize this;
L:245-256:
"In 0.9% NaCl solution, chloride ions stabilize the passive oxide layer by forming a protective barrier that mitigates surface degradation [48]. This stabilization reduces the CoF by limiting active material removal and preserving the surface integrity during sliding. Conversely, in Hanks' solution, the presence of phosphate and bicarbonate ions induces a competing interaction with the surface oxide layer. Phosphate ions can form weak complexes with metal oxides, while bicarbonate ions promote localized pH shifts, potentially destabilizing the oxide layer [49]. This destabilization leads to increased wear and higher friction coefficients. The tribochemical interactions are influenced by the ionic strength and concentration gradients, which alter the electrochemical potential of the oxide layer [50]. These results underscore the intricate interplay between the ionic composition of the surrounding environment and the mechanical and chemical processes driving tribocorrosive wear.”
Comment #3:
The paper does not mention the application of EIS, having focused mainly on anodic polarization and open circuit potential (OCP) methods to evaluate corrosion resistance. These tests are adequate, but EIS would allow a deeper analysis of the properties of the passive layer and the corrosion mechanisms. Therefore, I recommend including EIS as a complementary method, as it provides a more detailed view of the electrochemical interface parameters, such as charge transfer resistance and capacitance of the passive layer, presenting Nyquist and Bode plots and discussing the evolution of the resistance and stability of the passive layer in different simulated solutions.
Response:
We sincerely appreciate the reviewer’s suggestion to incorporate electrochemical impedance spectroscopy (EIS) as a complementary technique. Although incorporating EIS would undoubtedly enrich the analysis, the present study aims to assess the corrosion resistance and passive film behavior of the developed coatings using potentiodynamic polarization, which has been used in similar studies. We regret to inform the reviewer that we will not be able to perform additional EIS analysis at this time due to resource and time constraints, coupled with the scope of the experimental design.
Nonetheless, we have mentioned this in the revised manuscript as a recommendation for further research.
L:476-483:
“This work predominantly utilized potentiodynamic polarization to assess the corrosion resistance and passive film stability of the coatings, while recognizing the need of electrochemical impedance spectroscopy (EIS) for a more comprehensive understanding of the electrochemical behavior, providing insights into passive layer characteristics, such as charge transfer resistance and capacitance, which are critical in evaluating long-term performance under physiological conditions. While not within the scope of the present study, the incorporation of EIS will be prioritized in future research to attain a more thorough assessment of the coatings' performance.”

Reviewer 3 Report
Comments and Suggestions for Authors
The manuscript provides an innovative approach to enhancing the mechanical and chemical properties of 316L stainless steel. the manuscript requires major revisions to address issues related to experimental clarity, scientific rigor, and result interpretation. Below are detailed comments and critical questions for the authors:
1. The authors mention the use of hot pressing for coatings, but additional justification for the chosen parameters (e.g., temperature, pressure) and their optimization is necessary.
2. Why was 1050°C and 50 MPa chosen as sintering conditions? Have the authors investigated whether other parameters could yield superior coatings?
3. Some general sentences in the manuscript are not supported by proper references and occasionally cite irrelevant or low-relevance sources. For example, the statement, "Elements such as Nb, Ta, and Mo, alloyed with Ti, form oxides like Nb₂O₅, Ta₂O₅, and MoO₃ in addition to TiO₂. These oxides enhance the stability of the passive film layer in corrosive environments," on lines 62–64 requires proper references. Suggested references include https://doi.org/10.1038/s41598-023-29553-5 and https://doi.org/10.1016/j.bioactmat.2020.04.014. The authors should address this to ensure scientific accuracy and credibility.
4. Discuss discrepancies in wear and corrosion behavior across different simulated environments with greater depth.
5. Address why 316L/Ti showed higher corrosion resistance in 0.9% NaCl but not in Hanks' solution.
6. Did the vacuum environment influence diffusion kinetics? Could this have led to a specific type of intermetallic formation?
7. Given the known toxicity of vanadium in Ti6Al4V alloys, how do the authors address potential biocompatibility concerns?
8. How does the thickness and continuity of the oxide layer differ between pure Ti and Ti6Al4V coatings, and how does this impact their electrochemical behavior?
9. The friction coefficient behavior in NaCl versus Hanks’ solution was notably different. How do these findings compare with clinical conditions in load-bearing implants?
10. How does the observed wear debris interact with surrounding biological tissues? Have cytotoxicity or ion-release studies been conducted?
11. How does the performance of these coatings compare to other advanced coatings like TiN or DLC (Diamond-Like Carbon) in similar environments?
12. The y-axis of the Tafel plots presented in Figures 7(a) and 7(b) should include the unit "V vs. reference electrode" (e.g., SCE or Ag/AgCl) for clarity and standardization. This is essential for readers to accurately interpret the corrosion potential values and compare them with other studies.
Comments on the Quality of English Language1. Improve grammatical consistency and use technical terminology more precisely.
2. Figures and tables should be self-explanatory; some lack sufficient detail in captions.
Author Response
The manuscript provides an innovative approach to enhancing the mechanical and chemical properties of 316L stainless steel. the manuscript requires major revisions to address issues related to experimental clarity, scientific rigor, and result interpretation. Below are detailed comments and critical questions for the authors:
Response: We thank the Reviewer for addressing the issues with our manuscript. We hope that, following these changes, the quality of our manuscript would conform to your standards
Comment #1: The authors mention the use of hot pressing for coatings, but additional justification for the chosen parameters (e.g., temperature, pressure) and their optimization is necessary
Response: We have provided a detailed justification for the chosen parameters in the Materials and Methods section. The sentences provided below are related to the topic.
L:121-133: "Before selecting production parameters, preliminary production studies were conducted to optimize the processing conditions. Temperature optimization was a primary focus in these studies, while the applied pressure was determined based on the load-bearing capacity of graphite dies during sintering. The optimum parameters for coating quality were identified as 1050°C and 40 MPa. Temperatures below 1050°C resulted in insufficient bonding between the coating and the substrate, with pores detected at the inter-face. Consequently, the samples were produced at 1050°C under a constant uniaxial pressure of 40 MPa. Simultaneous sintering of 316L stainless steel, pure Ti, and Ti6Al4V alloy powders was performed using a DIEX VS50 pressure-assisted sintering system in a 10⁻⁴ mbar vacuum atmosphere to prevent oxidation. Following sintering, in-situ-formed pure Ti and Ti6Al4V coatings, approximately 1000 µm thick, were produced on 316L substrates approximately 3000 µm in thickness. Cylindrical samples with a diameter of 40 mm and a thickness of 4 mm were ultimately produced.”
Comment #2: Why was 1050°C and 50 MPa chosen as sintering conditions? Have the authors investigated whether other parameters could yield superior coatings?
Response: Firstly, the productions were conducted under a constant pressure of 40 MPa, rather than 50 MPa. This discrepancy was due to a typographical error, as the pressure was incorrectly stated as 50 MPa in the abstract section, while correctly noted as 40 MPa in the materials and methods section. The abstract section has now been updated to reflect the correct value of 40 MPa. Our response to your question is also included in the text, which justifies the selection of the processing parameters.
L:121-128: “Before selecting production parameters, preliminary production studies were conducted to optimize the processing conditions. Temperature optimization was a primary focus in these studies, while the applied pressure was determined based on the load-bearing capacity of graphite dies during sintering. The optimum parameters for coating quality were identified as 1050°C and 40 MPa. Temperatures below 1050°C resulted in insufficient bonding between the coating and the substrate, with pores detected at the interface. Consequently, the samples were produced at 1050°C under a constant uniaxial pressure of 40 MPa.”
Comment #3: Some general sentences in the manuscript are not supported by proper references and occasionally cite irrelevant or low-relevance sources. For example, the statement, "Elements such as Nb, Ta, and Mo, alloyed with Ti, form oxides like Nb₂O₅, Ta₂O₅, and MoO₃ in addition to TiO₂. These oxides enhance the stability of the passive film layer in corrosive environments," on lines 62–64 requires proper references.
Suggested references include https://doi.org/10.1038/s41598-023-29553-5 and https://doi.org/10.1016/j.bioactmat.2020.04.014. The authors should address this to ensure scientific accuracy and credibility.
Response: Thank you for your suggestion. We have double-checked the citations throughout the manuscript and updated the citations where necessary. For response to your specific suggestion, we have included the suggested references.
L:66-67: These oxides enhance the stability of the passive film layer in corrosive environments [21-25].
- Bordbar-Khiabani, A.; Gasik, M. Electrochemical and biological characterization of Ti–Nb–Zr–Si alloy for orthopedic applications. Rep. 2023, 13, 2312.
- Li, B.Q.; Xie, R.Z.; Lu, X. Microstructure, mechanical property and corrosion behavior of porous Ti–Ta–Nb–Zr. Mater. 2020, 5, 564-568.
Comment #4: Discuss discrepancies in wear and corrosion behavior across different simulated environments with greater depth
Response: As suggested, a more detailed analysis of the influence of electrolyte differences on wear and corrosion properties has been included.
L:368-386: “In Hanks' solution, the buffering effect and the presence of multivalent ions, such as phosphate and bicarbonate, introduce complex interactions with the passive oxide layer [63]. Phosphate ions can compete with chloride ions, altering the adsorption and dissolution dynamics at the metal interface. Bicarbonate ions further modify the local pH and potentially destabilize the oxide layer, increasing wear rates and a higher likelihood of localized corrosion [64]. In contrast, 0.9% NaCl, with its relatively simple ionic composition, primarily facilitates chloride-induced breakdown of the passive layer. However, this simpler environment allows for a more stable oxide layer under lower mechanical stresses, resulting in comparatively lower wear and corrosion rates. Differences in mechanical and electrochemical synergies between the two solutions play a significant role. In Hanks' solution, the simultaneous mechanical action and chemical reactivity exacerbate tribocorrosive wear, particularly in the Ti6Al4V coating, due to its sensitivity to multivalent ion interactions. The pure Ti coating demonstrates superior corrosion resistance in Hanks' solution, potentially due to forming a more adherent and stable oxide layer. In 0.9% NaCl, both coatings exhibit comparable corrosion resistance, suggesting a dominant role of chloride ions in determining the passive film's stability. These observations highlight the intricate interplay between chemical composition, pH buffering capacity, and tribological conditions in driving the material degradation processes in different electrolytes.”
Comment #5: Address why 316L/Ti showed higher corrosion resistance in 0.9% NaCl but not in Hanks' solution.
Response: This observation can be attributed to the competitive adsorption and complex ion interactions occurring in Hanks' solution, which differ from the stabilizing effect of chloride ions in NaCl on the TiO₂ passive layer. The revised manuscript has now emphasized this according to your suggestions. Please see our response to your previous comment.
Comment #6: Did the vacuum environment influence diffusion kinetics? Could this have led to a specific type of intermetallic formation?
Response: Although the study does not aim to examine or discuss the influences of processing conditions on the diffusion kinetics and formation of intermetallic. Please find below our comments to your question. We would like to inform you that we have not included these explanations in the revised manuscript as this would be beyond the scope of the paper.
A vacuum environment significantly reduces the presence of oxygen and other atmospheric gases, preventing the formation of oxide layers on the surfaces of the coating material and the substrate. Since oxide layers act as barriers to diffusion, the vacuum facilitates atom migration across the interface, thereby enhancing diffusion kinetics. In contrast, if the process were conducted in an atmospheric environment, oxide layers, such as TiO₂, could form on the surfaces of Ti or Ti6Al4V, potentially hindering the diffusion process. Similarly, the passive oxide layer on the 316L substrate could grow, further impeding the diffusion interface. Moreover, under vacuum conditions, the diffusion of gaseous contaminants, such as oxygen, nitrogen, and carbon, into the substrate and coating material is prevented, ensuring a clean and unimpeded diffusion pathway. The vacuum also reduces surface energy, enabling easier atomic movement. This is particularly beneficial at elevated temperatures, promoting atomic bonding and accelerating solid-state diffusion. On the other hand, the vacuum atmosphere facilitates the formation of intermetallic phases such as FeTi and Fe₂Ti at the 316L/Ti and 316L/Ti6Al4V interfaces by preventing oxidation and eliminating impurities. Under vacuum conditions, the combination of high temperature and pressure enhances the diffusion of Fe and Ti atoms, thereby promoting the growth of these intermetallic phases. Overall, the vacuum environment enhances diffusion kinetics, improves the interfacial bonding between the coating and the substrate, and minimizes the effects of impurities. In contrast, atmospheric conditions would likely result in oxide layer formation and other contaminants, slowing diffusion kinetics, weakening the interfacial bond, and adversely affecting mechanical properties.
Comment #7: Given the known toxicity of vanadium in Ti6Al4V alloys, how do the authors address potential biocompatibility concerns?
Response: We appreciate the reviewer’s comment regarding potential biocompatibility concerns related to vanadium in Ti6Al4V alloys. While it is acknowledged that vanadium can exhibit cytotoxicity under certain conditions, it is important to highlight that Ti6Al4V, remains one of the most widely used metallic biomaterials in clinical applications, including orthopedic implants, dental devices, and surgical instruments. Compared to the potential of excessive ion release from stainless steel implants and wear debris generated due to the relatively poor resistance of stainless steel, Ti-based coatings could provide higher biocompatibility while this should be studied through in-vitro and in-vivo testing. Thanks to your suggestions, this has been emphasized as a future work in the conclusion section.
L:466-467: Future studies should be conducted to assess the biocompatibility and cytotoxicity of these coatings for clinical application.
Comment #8: How does the thickness and continuity of the oxide layer differ between pure Ti and Ti6Al4V coatings, and how does this impact their electrochemical behavior?
Response: In response to your question, we have clarified this in the revised manuscript.
L:396-422: “The oxide layer on pure titanium is typically thicker and more stable compared to Ti6Al4V. Under normal physiological conditions, it forms a stable TiO₂ layer that can grow up to several nanometers in thickness, typically ranging from 1 to 10 nm [66, 67]. This oxide layer forms immediately upon exposure to air or oxygen-containing environments and serves as a protective barrier against further oxidation. Generally, the oxide layer on pure titanium is continuous and uniform, providing excellent corrosion resistance. In contrast, the oxide layer on Ti6Al4V is usually thinner, ranging from 1 to 5 nm [68]. the presence of Al and V in the alloy influences the oxide structure. While a TiO₂ layer still forms, it is often accompanied by additional oxide phases, such as Al₂O₃ and V₂O₅, which can affect the overall properties of the oxide layer [69]. However, the oxide layer on Ti6Al4V may not be as continuous as that on pure titanium. The alloy’s microstructure can lead to localized variations in the formation of the oxide layer. For example, Al contributes to a more compact and protective oxide layer, whereas vanadium may introduce discontinuities, increasing the potential for localized breakdown and making the oxide layer more prone to damage. Consequently, Ti6Al4V exhibits slightly lower corrosion resistance than pure titanium, especially in environments where localized breakdown of the oxide layer is likely. Additionally, the presence of V and Al in the oxide layer can contribute to galvanic corrosion when the oxide layer is disrupted due to vanadium’s higher electrochemical activity compared to titanium [70]. As a result, the electrochemical behavior of Ti6Al4V may demonstrate higher susceptibility to localized corrosion and lower long-term stability when com-pared to pure titanium. The polarization tests conducted within the range of -250 mV to +250 mV vs. SCE have provided valuable insights into the corrosion resistance and passive film stability of the developed coatings. It is significant to emphasize that ex-tending the polarization range to elevated potentials may provide further insights into the transition from continuous anodic dissolution to passivation, along with the mate-rial's behavior in more aggressive electrochemical conditions.”
Comment #9: The friction coefficient behavior in NaCl versus Hanks’ solution was notably different. How do these findings compare with clinical conditions in load-bearing implants?
Response: We have expanded the discussion in the Results section to explore the influence of chemical components in the simulated solutions on tribocorrosive mechanisms. For instance, we included;
L: 245-256: “In 0.9% NaCl solution, chloride ions stabilize the passive oxide layer by forming a protective barrier that mitigates surface degradation [48]. This stabilization reduces the CoF by limiting active material removal and preserving the surface integrity during sliding. Conversely, in Hanks' solution, the presence of phosphate and bicarbonate ions induces a competing interaction with the surface oxide layer. Phosphate ions can form weak complexes with metal oxides, while bicarbonate ions promote localized pH shifts, potentially destabilizing the oxide layer [49]. This destabilization leads to in-creased wear and higher friction coefficients. The tribochemical interactions are influenced by the ionic strength and concentration gradients, which alter the electrochemical potential of the oxide layer [50]. These results underscore the intricate interplay between the ionic composition of the surrounding environment and the mechanical and chemical processes driving tribocorrosive wear.”
Comment #10: How does the observed wear debris interact with surrounding biological tissues? Have cytotoxicity or ion-release studies been conducted?
Response: We have not studied the interaction of produced wear debris with surrounding biological tissues as this would be beyond the scope of the paper.
Comment #11: How does the performance of these coatings compare to other advanced coatings like TiN or DLC (Diamond-Like Carbon) in similar environments?
Response: We have added additional text into the introduction to address your question.
L:69–83: “Increased passive film thickness correlates with enhanced corrosion resistance. On the other hand, diamond-like carbon (DLC) and TiN have emerged in recent years as potential materials for biomaterials due to their high hardness, low friction coefficient, excellent wear and corrosion resistance, chemical inertness, and high electrical resistance [29, 30]. However, significant debates persist regarding the effectiveness of these coatings for biomedical applications. For instance, clinical trials of DLC-coated vascular stents have revealed that DLC coatings do not provide a substantial improvement in restenosis rates compared to uncoated stents [31]. Furthermore, the production processes for coatings like TiN and DLC are typically more complex and expensive [32]. Lastly, a ten-year follow-up study of DLC-coated artificial hip joints demonstrated that the failure rate of DLC-coated Ti6Al4V femoral heads was significantly higher than that of alumina femoral heads [33]. Therefore, although TiN and DLC coatings are harder and more wear-resistant, the inherent properties of pure Ti and Ti6Al4V make them more suitable for biocompatible and long-lasting implants.”
Comment #12: The y-axis of the Tafel plots presented in Figures 7(a) and 7(b) should include the unit "V vs. reference electrode" (e.g., SCE or Ag/AgCl) for clarity and standardization. This is essential for readers to accurately interpret the corrosion potential values and compare them with other studies
Response: Following your suggestion, the y-axis of Figure 7 has been revised, and the figure has been updated.
Figure 7. Tafel extrapolation curves of 316L, 316L/Pure Ti, and 316L/Ti6Al4V in (a) 0.9% NaCl and (b) Hanks’ solution.
Comment #13: Improve grammatical consistency and use technical terminology more precisely
Response: Thank you for your valuable feedback. We have made the necessary revisions to improve grammatical consistency and use technical terminology more accurately. We have carefully reviewed the text, taking your comments into account, and made significant corrections in this regard.
Comment #14: Figures and tables should be self-explanatory; some lack sufficient detail in captions.
Response: The figure captions have been revised according to your suggestions. The updated titles have been highlighted in yellow in the revised manuscript.

Reviewer 4 Report
Comments and Suggestions for Authors
For a study in the evaluation of coatings, especially in the biomedical area, the electrochemical impedance spectroscopy technique should be used to complement the analysis of the coating behavior. On the other hand, the polarization range (-250 mV to +250 mV vs. SCE at a rate of 1 mV/s ) used in the present study is insufficient to evaluate the behavior in the anodic branch in the polarization curves, at least it should be polarized up to +600 or +700 mV in order to determine in a complete way the behaviors of both continuous anodic dissolution and passivation, the polarization curves presented by the authors are not complete, in the anodic branch it clearly appeared that there is a tendency to change the behavior at a higher polarization range.
The authors are recommended to revise the experimental part in order to present useful and reliable results.

Author Response
Comment #1: For a study in the evaluation of coatings, especially in the biomedical area, the electrochemical impedance spectroscopy technique should be used to complement the analysis of the coating behavior.
Response: We sincerely appreciate the reviewer’s suggestion to incorporate electrochemical impedance spectroscopy (EIS) as a complementary technique. Although incorporating EIS would undoubtedly enrich the analysis, the present study aims to assess the corrosion resistance and passive film behavior of the developed coatings using potentiodynamic polarization, which has been used in similar studies. We regret to inform the reviewer that we will not be able to perform additional EIS analysis at this time due to resource and time constraints, coupled with the scope of the experimental design.
Nonetheless, we have mentioned this in the revised manuscript as a recommendation for further research.
L:476-484: “This work predominantly utilized potentiodynamic polarization to assess the corrosion resistance and passive film stability of the coatings, while recognizing the need of electrochemical impedance spectroscopy (EIS) for a more comprehensive understanding of the electrochemical behavior, providing insights into passive layer characteristics, such as charge transfer resistance and capacitance, which are critical in evaluating long-term performance under physiological conditions. While not within the scope of the present study, the incorporation of EIS will be prioritized in future research to attain a more thorough assessment of the coatings' performance.”
Comment #2: The polarization range (-250 mV to +250 mV vs. SCE at a rate of 1 mV/s) used in the present study is insufficient to evaluate the behavior in the anodic branch in the polarization curves, at least it should be polarized up to +600 or +700 mV in order to determine in a complete way the behaviors of both continuous anodic dissolution and passivation, the polarization curves presented by the authors are not complete, in the anodic branch it clearly appeared that there is a tendency to change the behavior at a higher polarization range.
Response: Thank you for your valuable feedback regarding the polarization range used in the study. We recognize that extending the potential range up to +600 or +700 mV, as suggested, could provide additional insights into the anodic dissolution and passivation behavior. We selected the polarization range (-250 mV to +250 mV vs. SCE) to align with the conditions relevant to biological environments. The primary focus of this work was to investigate corrosion resistance and passive film stability within these limits. Currently, we are unable to extend the polarization tests due to limited availability of sample sets and practical constraints. To address this feedback and enhance clarity, we have included the following modifications in the manuscript:
L:169-173: The polarization range utilized in this study (-250 mV to +250 mV vs. SCE) was selected to reflect the conditions most relevant to the biological environments in which the materials are expected to perform. This range was specifically chosen to evaluate the passive film formation and initial corrosion resistance behavior under simulated conditions without risking sample integrity.
L:416-422: The polarization tests conducted within the range of -250 mV to +250 mV vs. SCE have provided valuable insights into the corrosion resistance and passive film stability of the developed coatings. It is significant to emphasize that extending the polarization range to elevated potentials may provide further insights into the transition from continuous anodic dissolution to passivation, along with the material's behavior in more aggressive electrochemical conditions.
L:468-475: The polarization range used in this investigation (-250 mV to +250 mV vs. SCE) was chosen to reflect the biological settings in which approximate passive film development and initial corrosion resistance without compromising sample integrity can be observed. However, increasing the polarization range to higher potentials could reveal more about the transition from continuous anodic dissolution to passivation and the material's reaction in more aggressive electrochemical conditions. Thus, Thus, future research will include an expanded potential range to better understand these coatings' anodic behavior under extended electrochemical conditions.

Round 2
Reviewer 2 Report
Comments and Suggestions for Authors
Dear Author,
The paper has been improved, and it can be accepted as is.
Author Response
Dear Author,
The paper has been improved, and it can be accepted as is.
Response:
Thank you for recognizing the enhancements made to the paper and for your final recommendation regarding its publication.

Reviewer 3 Report
Comments and Suggestions for Authors
Accept in the present form.
Author Response
Accept in the present form.
Response: Thank you for recognizing the enhancements made to the paper and for your final recommendation regarding its publication.

Reviewer 4 Report
Comments and Suggestions for Authors
The potentiodynamic polarization technique is an electrochemical test method in which the corrosion process that will present a coating-substrate system is accelerated in the case of this manuscript, and depends on the range of polarization to obtain the complete behavior of this phenomenon. Medical implants remain in the body the necessary time for the affected tissue or limb to be reestablished, so the corrosion mechanism continues acting during all the time that the implant is present in the body, either temporary or permanent, that is why it must be guaranteed that the results in this type of electrochemical tests are performed in a complete way, considering the appropriate polarization ranges to evaluate the entire electrochemical cycle in both the cathodic and anodic branches. As well as being complemented with EIS studies.
Therefore, it is recommended that the authors clarify both in the introduction and in the conclusions that the results presented in this study form part of the beginning or are the first part of the corresponding study. Especially if it is not considered to complement them with the performance of the tests and parameters suggested in the first review.
Author Response
Comment #1: The potentiodynamic polarization technique is an electrochemical test method in which the corrosion process that will present a coating-substrate system is accelerated in the case of this manuscript, and depends on the range of polarization to obtain the complete behavior of this phenomenon. Medical implants remain in the body the necessary time for the affected tissue or limb to be reestablished, so the corrosion mechanism continues acting during all the time that the implant is present in the body, either temporary or permanent, that is why it must be guaranteed that the results in this type of electrochemical tests are performed in a complete way, considering the appropriate polarization ranges to evaluate the entire electrochemical cycle in both the cathodic and anodic branches. As well as being complemented with EIS studies.
Therefore, it is recommended that the authors clarify both in the introduction and in the conclusions that the results presented in this study form part of the beginning or are the first part of the corresponding study. Especially if it is not considered to complement them with the performance of the tests and parameters suggested in the first review.
Response: We sincerely appreciate and agree with the reviewer’s concluding remarks. As requested, we have clarified the suggested comments in both the introduction and conclusion sections.
L:107-112:
The results given in this study constitute the initial part of the associated research. For a better understanding of electrochemical behavior, electrochemical impedance spectroscopy (EIS) and research across a wider polarization range will be used to study passive layer characteristics like charge transfer resistance and capacitance, which are crucial for assessing the long-term performance of the developed coatings under physiological conditions.
L:475-478:
The findings presented in this study represent the preliminary phase of the associated research. Thus, future research will include an expanded potential range to better understand these coatings' anodic behavior under extended electrochemical conditions.
L:484-485:
The incorporation of EIS will be prioritized in future research to attain a more thorough assessment of the coatings' performance.
